# Tracking karyotype dynamics by flow cytometry reveals de novo chromosome duplications in laboratory cultures of *Macrostomum lignano*

Stijn Mouton*, Lisa Glazenburg and Eugene Berezikov*

## ABSTRACT

The flatworm *Macrostomum lignano* is a versatile invertebrate model organism with a growing molecular toolbox. Genome assembly and detailed karyotyping revealed that *M. lignano* is a hidden polyploid species with a recent whole-genome duplication. Its karyotype consists of six small and two large chromosomes (*2n=8*), with the large chromosomes originating from the fusion of duplicated ancestral chromosomes. However, *2n=9* and *2n=10* karyotypes with duplicated large chromosomes were also observed in animals from both laboratory cultures and field samples, prompting us to further investigate this phenomenon. To this end, we optimized a flow cytometric approach that enables easy and rapid studies of tens or even hundreds of animals simultaneously to gain insight into the karyotype polymorphisms present in a culture and consistently tracked karyotype dynamics in multiple cultures over a period of 26 months. We demonstrate that *de novo* duplications of the large chromosome in *M. lignano* can spontaneously appear under laboratory conditions and can become dominant in laboratory cultures. Since uncontrolled chromosomal duplications can complicate genetic studies in laboratory model organisms, we propose an approach to easily control the karyotype of experimental cultures by regularly karyotyping *M. lignano* subcultures using flow cytometry and replacing cultures with *de novo* chromosome duplications as needed.

KEY WORDS: *Macrostomum lignano*, Karyotype dynamics, Aneuploidy, Genome duplication, Flow cytometry, Flatworm

## INTRODUCTION

*Macrostomum lignano* is a free-living marine flatworm that is approximately 1 mm long (Fig. 1A). While this transparent worm might seem unremarkable at first, it is gaining traction as a model organism to study diverse biological questions (Wudarski et al., 2020), including regeneration (Hall et al., 2024; Kibet et al., 2025), bio-adhesion (Wunderer et al., 2019), and male fertility (Weber et al., 2020). A key factor in *M. lignano*'s appeal as a model system is its ease of culturing under laboratory conditions. Large cultures can be easily generated and maintained by keeping worms in Petri dishes with artificial sea water (ASW) and diatoms as a food source.

*M. lignano* is a non-self-fertilizing hermaphrodite that reproduces exclusively in a sexual manner (Schärer and Ladurner, 2003). The worms lay abundant single-cell fertilized eggs, and the egg-to-egg generation time is 2-3 weeks (Morris et al., 2004; Wudarski et al., 2019). Notably, the eggs can be microinjected, which has enabled the development of robust transgenesis methods (Wudarski et al., 2017), and *M. lignano* remains the only flatworm species in which transgenesis is currently possible. To support functional studies, transcriptome and genome assemblies have been created and are publicly available at the Flatworms and Acoels Genome Browser.

Assembling the genome (Wudarski et al., 2017) and detailed karyotyping (Zadesenets et al., 2016, 2017a,b) revealed that *M. lignano* is a hidden polyploid species. The modern genome of this species was formed through a recent whole-genome duplication, followed by re-diploidization, during which all ancestral chromosomes fused into one large chromosome (Zadesenets et al., 2020, 2023). This resulted in a *2n=8* karyotype with one pair of large and three pairs of small metacentric chromosomes (Fig. 1B) (Egger and Ishida, 2005). However, this karyotype appears unstable. Substantial karyotype polymorphisms were observed between individual *M. lignano* worms, both in laboratory cultures and in field-collected specimens. The main variation is aneuploidy of the large chromosome, resulting in *2n=9* and *2n=10* karyotypes (Fig. 1B) (Zadesenets et al., 2016). A gain or loss of small chromosomes has only been observed in some rare cases (Zadesenets et al., 2017b). Interestingly, tri- and tetrasomy of the large chromosome do not adversely affect worm morphology and fertility. On the contrary, aneuploid karyotypes are not only tolerated but are increasingly prevalent after long-term culturing under laboratory conditions (Zadesenets et al., 2016, 2020). To date, it remains unclear whether this increasing number of aneuploid worms is due to the fitness advantages of a few aneuploid worms present when starting laboratory cultures from field samples or if *de novo* duplications of the large chromosome can occur in the laboratory (Zadesenets et al., 2016, 2020).

By consistently tracking the karyotype dynamics of laboratory cultures for 26 months by means of flow cytometry, we here unambiguously demonstrated that *de novo* chromosome duplications occur under laboratory conditions.

## RESULTS

### Creating a wild-type *M. lignano* culture without karyotype polymorphisms

To study whether chromosome duplications can occur *de novo* in laboratory cultures, we first took several steps to obtain a culture without any karyotype polymorphisms, called NL12S. For this purpose, we made subcultures of the NL10 *M. lignano* culture by randomly selecting 30 juvenile worms and making 15 isolated pairs, of which 14 gave rise to starting subcultures. NL10 is a wild-type laboratory culture that was established from worms collected near

European Research Institute for the Biology of Ageing, University Medical Center Groningen, University of Groningen, Groningen 9700AD, The Netherlands.

*Authors for correspondence (s.m.mouton@umcg.nl; e.berezikov@umcg.nl)

S.M., 0000-0002-1123-6268; E.B., 0000-0002-1145-2884

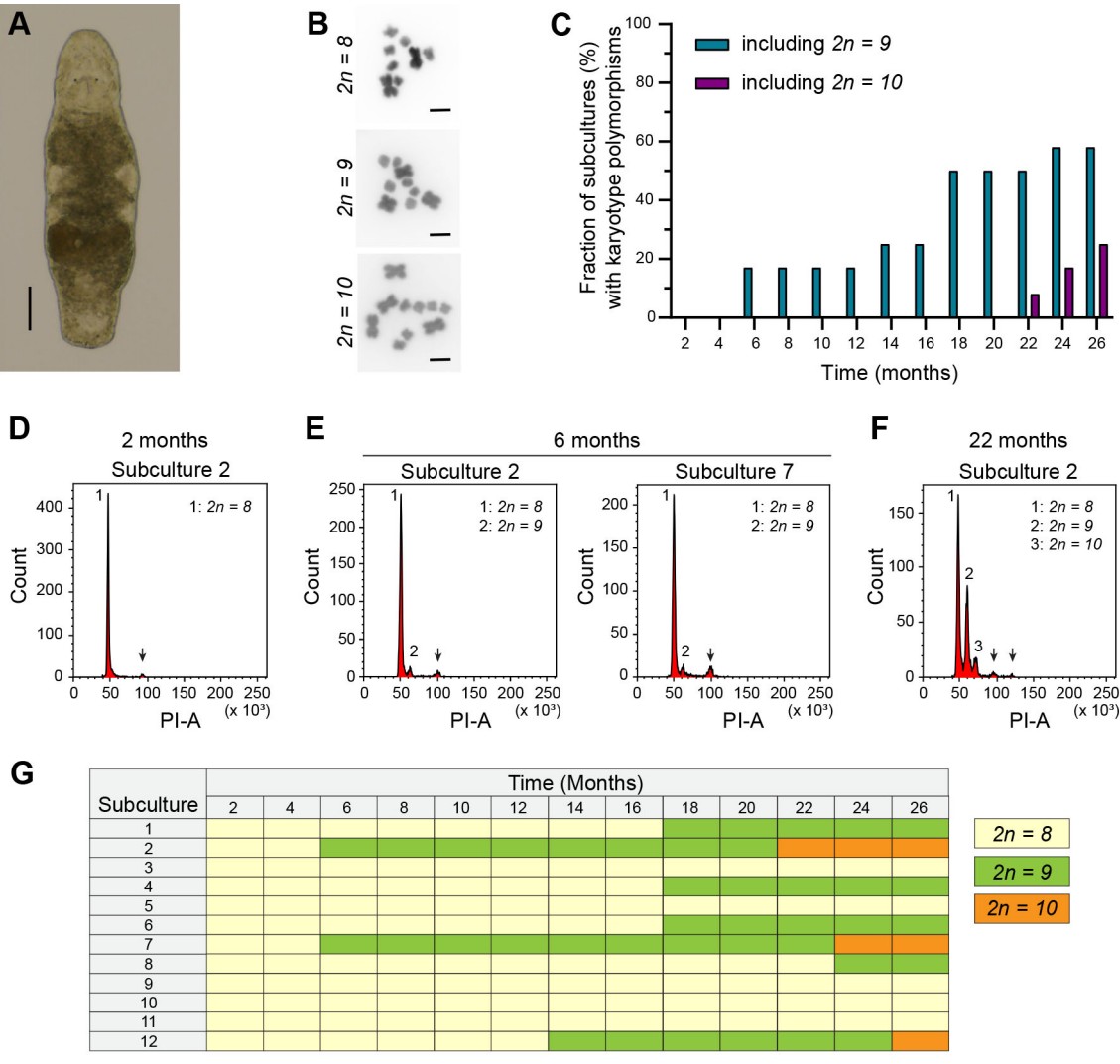

**Fig. 1. Karyotype evolution in laboratory cultures of *M. lignano*.** (A) Brightfield image of an adult worm. Scale bar: 100 μm. (B) Karyotype polymorphisms with an increasing number of large chromosomes that can be found in *M. lignano*. Scale bars: 10 μm. (C) Dynamics of karyotype polymorphisms present in 12 experimental subcultures over a period of 26 months. All subcultures started with only the *2n=8* karyotype. The bars indicate the percentage of subcultures that also contain the *2n=9* and *2n=10* karyotypes as a function of time. (D-F) Flow cytometric histograms based on Propidium Iodide area (PI-A), which visualize karyotype polymorphisms present in a culture. The different peaks (1-3) represent diploid cells with different karyotypes and thus different amounts of DNA. Arrows represent cells in the G2 and mitotic cell cycle phases that do not impact karyotype interpretation. (D) Example histogram representing the *2n=8* karyotype at the start of the experiment. (E) The first appearance of a second peak (2) arising due to the duplication of a large chromosome, thus representing a *2n=9* karyotype. (F) The first appearance of the *2n=10* karyotype (peak 3) at 22 months. (G) Visualization of the appearance of chromosome duplications in the 12 subcultures as a function of time.

Lignano, Italy, in 2014 (Wudarski et al., 2017) and was propagated in the laboratory for 3 years before we started this study. Flow cytometry-based karyotyping of these subcultures revealed that only six of them were characterized by a single *2n=8* karyotype (Fig. S1A). These six subcultures were further grown to include hundreds of worms and were analyzed again 6 weeks later. Only four of them showed a clear single peak and were thus characterized by a single *2n=8* karyotype (Fig. S1B). These subcultures were combined, renamed NL12S ('S' for 'single'), and further grown as a novel laboratory culture without chromosomal duplications.

## Chromosome duplications occur *de novo* in laboratory cultures of *M. lignano*

Starting with pairs of worms, 12 experimental subcultures of the NL12S culture were established and examined every 2 months for a period of 26 months. Flow cytometry-based karyotyping performed after 2 and 4 months confirmed that all subcultures initially contained only the *2n=8* karyotype (Fig. 1D; Fig. S2-S13). Already after 6 months, the first chromosome duplications, leading to a *2n=9* karyotype, were observed in subcultures 2 and 7 (Fig. 1C,E,G). During the first year of the experiment, these remained the only two subcultures, representing 17% of all studied cultures, in which chromosome duplication occurred (Fig. 1G; Fig. S2-S13). From 14 months onwards, chromosome duplications were also observed in other subcultures (Fig. 1C,G; Fig. S2-S13). By the end of the experiment at 26 months, seven subcultures (58%) included the *2n=9* karyotype (Fig. 1C,G; Fig. 2). The *2n=10* karyotype was first observed in subculture 2 at 22 months (Fig. 1C,F,G). By the 26 months timepoint, three subcultures (25%) included this karyotype polymorphism (Fig. 1C,G). At this time, the ratio of

karyotypes present in each subculture clearly varied, as shown in Fig. 2. It is worth noting that it is impossible to distinguish worms with different karyotypes by microscopy, suggesting that the karyotype has no clear impact on the morphology or motility of the worms. Interestingly, once a novel karyotype polymorphism with additional large chromosomes appeared, the number of worms in the subculture with this polymorphism always increased over time during our studies, as judged by the increased number of nuclei with a higher DNA content (Fig. S2-S13). Moreover, the karyotype dynamics of the experimental subcultures appeared to be unidirectional. Additional chromosomes can be gained but are never lost. These trends are illustrated by subculture 2 in Fig. 1D-F. Taken together, this further suggests that aneuploid worms could have fitness advantages and slowly outnumber worms with the original *2n=8* karyotype. As a consequence, the *2n=10* karyotype will become dominant, or even the only one, in laboratory cultures of *M. lignano* in the long run (Fig. S14).

### Controlling culture karyotypes

After the experiment, we maintained the subcultures with the single *2n=8* karyotype. Six years after the start of the experiment, only subculture 10 showed no additional karyotype polymorphisms. For experiments, we aim to maintain a large wild-type culture without chromosome polymorphisms (NL12S) at all times. To achieve this, we perform annual flow cytometry of the NL12S culture. When

chromosome duplications were observed, the culture was discarded, and a new NL12S culture was grown from subculture 10. To keep track of the origin of the NL12S culture, the year in which it was made was added to the name. Currently, we are working with NL12S23 and provide this culture to the community. If needed, novel subcultures can be established from NL12S23 to replace the culture.

## DISCUSSION

In this study, we characterized karyotype dynamics of *M. lignano* over a period of 26 months using a methodology based on flow cytometry of Propidium Iodide (PI)-labeled nuclei. The essence of this technique is that the intensity of PI fluorescence represents the amount of DNA in the nuclei, which increases with the duplication of large chromosomes, and therefore reflects the karyotype polymorphisms present in a culture. This flow cytometric approach has been previously used to estimate the genome size of multiple eukaryotic species by comparing samples of species with unknown genome sizes to control samples of species with known genome sizes (Hare and Johnston, 2011). During the genome sizing of *M. lignano* (Wudarski et al., 2017), we recognized the potential of this method for identifying karyotype variations in this species. Traditional metaphase chromosome preparations are essential for identifying different karyotype polymorphisms in a species. However, once these are known, the flow cytometric approach provides several advantages

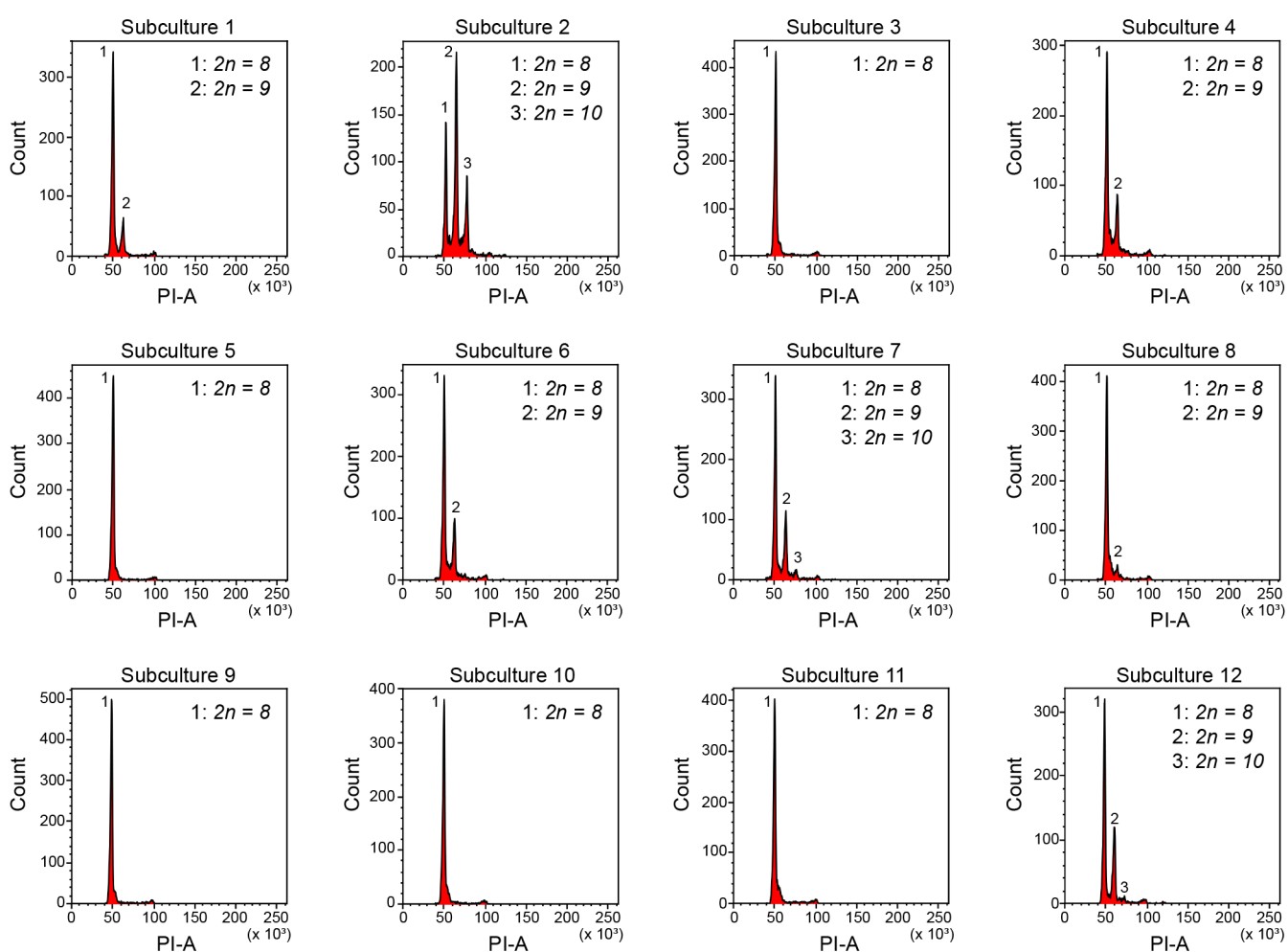

**Fig. 2. Karyotype polymorphisms present after 26 months.** Overview of the PI-A-based histograms of the 12 experimental subcultures at the end of the study after 26 months. The number of karyotype polymorphisms present and the ratio of nuclei between them varies considerably.

over the traditional method because it is easier, faster, less biased as it is less dependent on personal skills, and it provides information on a large number of nuclei, reflecting the karyotype polymorphisms of numerous worms in a laboratory culture.

Using this approach, we unambiguously demonstrated that *de novo* duplications of large chromosomes occur in laboratory cultures of *M. lignano*. Moreover, these duplications are not rare, as they occurred in seven of the 12 studied subcultures (58%) within 26 months. Within 1 year, two subcultures obtained a *2n=9* karyotype; therefore, we recommend that the *Macrostomum* community evaluate the karyotype of important laboratory cultures at least once a year.

The combination of flow cytometry-based karyotyping and the creation of subcultures starting from worm pairs provides a way to study karyotype dynamics and control culture karyotypes. Subcultures can be made at any time, but maintaining a small number of subcultures that are evaluated yearly makes it possible to quickly grow a large experimental culture with only the *2n=8* karyotype when required. Importantly, these tools are not only convenient for maintaining the *2n=8* karyotype but could also be used to accelerate the generation of cultures with increasing numbers of worms with karyotype polymorphisms and to obtain cultures with only *2n=9* or *2n=10* karyotypes. This is facilitated by the unidirectional nature of karyotype changes in laboratory cultures. As previously reported (Zadesenets et al., 2016), the number of aneuploid worms appears to increase over time in laboratory conditions. Moreover, although chromosomes can be gained, the loss of additional chromosomes has not yet been observed.

Taken together, these results suggest that aneuploidy is tolerated and can even result in fitness advantages in *M. lignano*. Tolerance to aneuploidy is not unique to *M. lignano* and has been previously described in other representatives of the flatworm genus *Macrostomum* and species from different taxa, including plants, fungi, and protozoa (Zadesenets and Rubtsov, 2025). Interestingly, polyploid organisms are more tolerant to aneuploidy, particularly those that have recently undergone whole-genome duplication (Zadesenets and Rubtsov, 2025). *M. lignano* falls within this category as it is a hidden polyploid with the large chromosome being formed by a fusion of all ancestral small chromosomes (Zadesenets et al., 2023). This unusual genomic composition of *M. lignano* makes it a special case where the concepts of aneuploidy and polyploidy merge because the duplication of the large chromosome also means another duplication of the ancestral whole genome. Whole-genome duplications have been reported in different invertebrate lineages, including rotifers, snails, nematodes, and arthropods (Au et al., 2025). However, despite invertebrates comprising more than 95% of all described animal species, the study of WGD events in invertebrate lineages is still in its infancy compared to that in plants, fungi, and vertebrates. *M. lignano* represents an interesting and convenient invertebrate model to perform research on the mechanisms of karyotype and genome evolution after a recent whole genome duplication event.

## MATERIALS AND METHODS
### *M. lignano* cultures
Worms were kept in Petri dishes with Guillard's f/2 medium (Anderson et al., 2005), a nutrient-enriched artificial sea water medium, at a salinity of 32‰. The dishes also contained pre-grown diatoms of the unicellular species *Nitzschia curvilineata* (SAG, Göttingen, Germany) as a food source. Petri dishes with worms and diatoms were maintained with a 14 h/10 h day/ night rhythm and a temperature of 20°C or 25°C. Increasing the temperature from 20°C to 25°C increases the fertility of worms and shortens the total generation time without inducing stress (Wudarski et al., 2019), which enables growing larger cultures in less time. Worms were transferred weekly or biweekly to new Petri dishes containing fresh medium and diatoms.

### Creating and maintaining experimental subcultures
Experimental subcultures of *M. lignano* were created by placing pairs of juvenile worms in separate wells of six-well plates containing f/2 medium and diatoms. Once they reached adulthood, the couples began reproducing, resulting in subcultures. All worms within a well were transferred weekly to new wells containing fresh media and food. When a well contained multiple adult worms, they were transferred to a Petri dish for further subculture growth. Most subcultures contained hundreds of worms within 2 months, and all worms were transferred weekly to new dishes with fresh medium and diatoms to provide *ad libitum* food. Because all worms were transferred, the subcultures included a mix of different generations of worms. Multi-well plates and Petri dishes with worms were maintained at 25°C with a 14 h/10 h day/night rhythm.

### Tracking karyotype evolution of *M. lignano* laboratory cultures
To study the *de novo* appearance of karyotype polymorphisms under laboratory conditions, 12 experimental subcultures of wild-type worms were created. These subcultures were maintained at 25°C and studied every 2 months for a total period of 26 months. The karyotype polymorphisms were analyzed using flow cytometry of PI-labelled single-nuclei suspensions of at least 100 worms. The first timepoint (2 months) deviates as it lacks for subcultures 3, 6, and 9, and the analysis of subcultures 1, 10, and 11 was performed with less than 100 worms. This is due to differences in the initial growth of the subcultures.

### Generating single-nuclei suspensions of *M. lignano*
To prepare a suspension of PI-labelled single nuclei, at least 100 worms were selected and starved for 24 h to eliminate residual diatoms present in the gut. The worms were then collected in an Eppendorf tube, and the f/2 medium was removed and replaced with 200 µl 1× Accutase (Sigma, A6964). After 15 min of incubation at room temperature, the medium containing worms was harshly pipetted up and down with a P200 pipette while visually inspecting it under a stereomicroscope. This was performed until the worms broke into fragments of variable sizes. The number of times one should pipette is variable depending on the initial size of the worms and the efficiency of removing the f/2 medium. After another 15 min of incubation, the fragments were dissociated into a single-cell suspension by harshly pipetting up and down with a P200 pipette. The number of times can vary depending on the previous steps but is sufficient when the solution is homogeneous and no pieces of tissue can be observed anymore. Then, 800 µl f/2 medium was added, and everything was mixed by pipetting up and down ten times with a P200 pipette. The single-cell suspension was then directly transferred to a 15 ml Falcon tube. The cells were pelleted by centrifugation at 1000 rpm for 5 min at 4°C using a centrifuge with a swing-bucket rotor. The supernatant was aspirated, and the cell pellet was resuspended in 1 ml nuclei isolation buffer (100 mM Tris-HCl pH 7.4, 154 mM NaCl, 1 mM $CaCl_2$, 0.5 mM $MgCl_2$, 0.02% BSA, 0.1% NP-40 in MilliQ water) with RNAse A (10 µg/ml) and PI (10 µg/ml) by pipetting up and down with a P200 pipette until no clumps were observed. It is worth noting that the cell pellet can be very small and difficult to observe when only 100 worms or fewer are used. In this case, sufficient supernatant (until above the expected location of the pellet) should be left before adding the nuclei isolation buffer to avoid losing the pellet. The nuclei suspension was then immediately passed through a 45 µm filter into the tubes for flow cytometry using a P200 pipette. The samples were incubated in the dark for 15 min on ice.

### Flow cytometry-based karyotyping
The nuclei suspensions were examined using a BD FacsCanto II Cell Analyzer. Flow cytometry enables the rapid investigation of large numbers of nuclei and, thus, many worms in a culture. As PI intercalates into the major groove of DNA, its fluorescence intensity reflects the amount of DNA present in the nuclei. Therefore, duplications of the large chromosome of *M. lignano* lead to clear shifts in PI intensity, visualizing the karyotype polymorphisms present in the studied culture as separate peaks (Fig. 3C). The shift in PI intensity is probably not enough to identify a change in the number of small chromosomes, but these changes are rare (Zadesenets et al., 2017b) and therefore not the focus of this study. The analysis was

performed using a straightforward gating strategy in the Kaluza Analysis Software (Fig. 3A; Fig. S15). Initially, PI-labelled nuclei were selected to distinguish intact nuclei from the cellular debris. This was performed by gating PI-positive events (Fig. 3A). For at least the first experiments, it is recommended to also measure a non-labelled sample to visualize PI-negative nuclei and debris. A selection gate can then be drawn above the negative events. When the PI-labelled sample is measured, the PI-positive nuclei should be easily recognized and situated within the drawn gate. The debris will be negative for PI and situated under the gate. Flatworm cell and nuclei suspensions contain large amounts of cellular debris, and it can be difficult to distinguish debris fragments from nuclei based on size (FSC) and internal complexity (SSC) because of the small size of *M. lignano* cells and nuclei (Fig. 3B). For this reason, we did not include the SSC versus FSC graph into the gating strategy. As the second step, the PI-positive nuclei were visualized in a PI-Width (W) versus PI-Area (A) plot to discriminate the nuclei aggregates and doublets. In this plot, single nuclei are represented by a slightly tilted horizontal line, while doublets and nuclei clusters will have larger PI-W values. To select the single nuclei, a gate is drawn as shown in the second plot of Fig. 3A, which selects the events with the lowest PI-W values. Finally, a histogram of the PI-A was used to visualize the fluorescence intensity of the PI labelling of the DNA, which reflects the

karyotypes present in the culture (Fig. 3A,C; Fig. S15). In most cases, we made final adjustments to the laser power to situate the *2n=8* peak at the 50 value of the X-axis, as shown in the third plot of Fig. 3A. This facilitates the comparison of results obtained at different time points. In general, a large peak of diploid cells in the G1 phase of the cell cycle and a very small peak of cells with double the amount of DNA in the G2 and Mitotic cell cycle phases can be observed in the histogram (Fig. 1D-F; Fig. 3A; Fig. S15). The peak of diploid cells is used to assess which karyotypes are present in the culture, as duplications of the large chromosome will appear as additional peaks.

### Karyotyping with metaphase chromosome preparations

Chromosome slides were prepared as previously described (Wudarski et al., 2017; Zadesenets et al., 2016). First, the bodies of worms were amputated, and the head fragments were left to regenerate for 48 h to increase the number of mitotic cells. The regenerating heads were collected and treated with 0.2% colchicine (Sigma, C9754-100 mg) in f/2 medium for 4 h at room temperature (RT). This step arrests mitosis at the metaphase. The head fragments were then treated with hypotonic 0.2% KCl solution for 1 h at RT. Next, the fragments were placed on clean glass slides in a mixture of 4:3:3 distilled water:ethanol:glacial acetic acid and macerated into small pieces

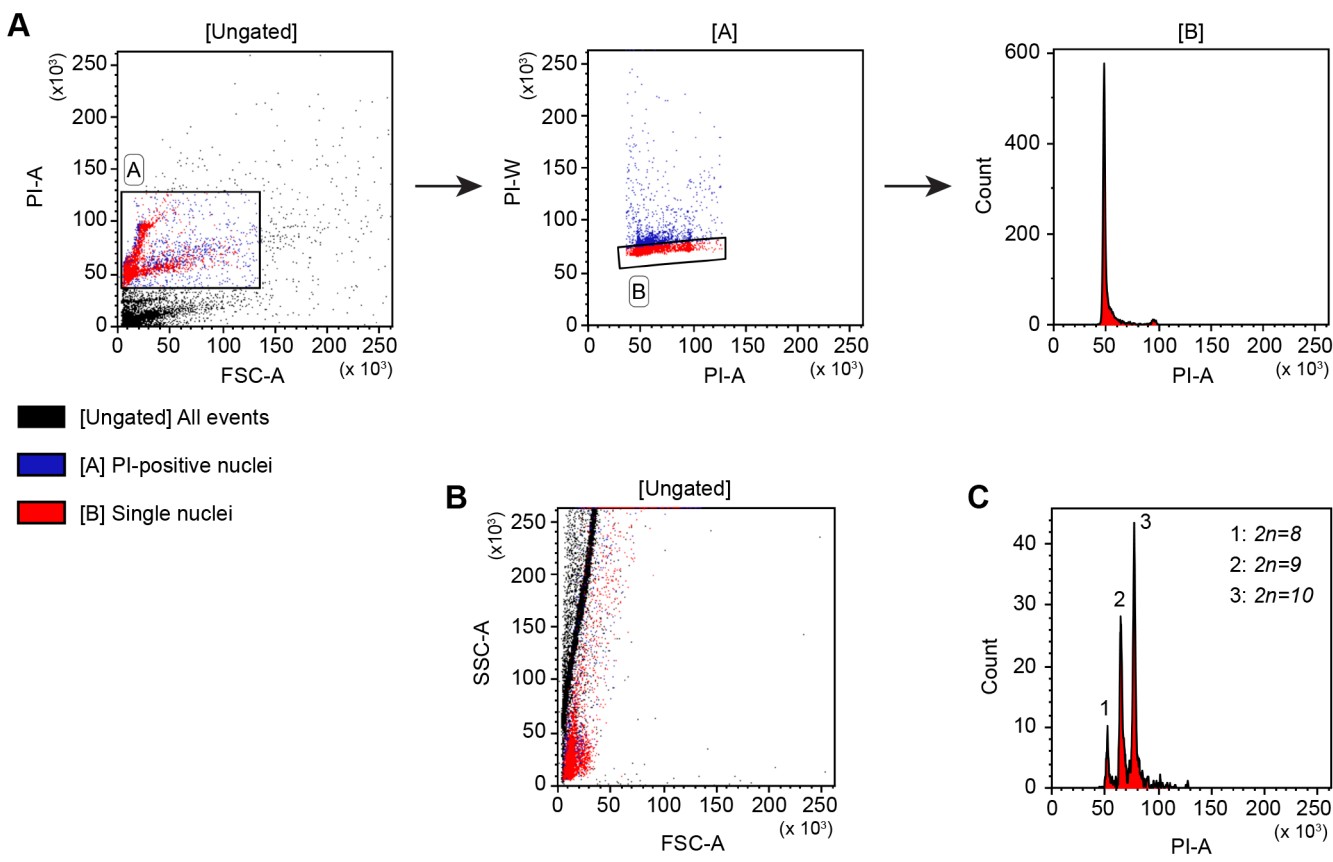

**Fig. 3. Flow cytometric approach to identify karyotype polymorphisms in *M. lignano* cultures.** (A) Gating strategy to identify karyotype polymorphisms present in a culture. 'FSC-A' represents the size, 'PI-A' represents the total PI fluorescence, 'PI-W' represents the time a nuclei spends while passing through the laser, and 'Count' represents the number of visualized nuclei. The first plot shows all the measured particles (black). Gate A represents the selection of all particles with PI labelling and thus represents the nuclei (in blue). The second plot shows only the nuclei selected in gate A (blue). Gate B selects the single nuclei (in red) and discriminates the clusters. The third plot is a histogram showing only the single nuclei (red). The large peak at the 50-value represents all diploid cells with a *2n=8* karyotype. The small peak at the 100-value represents nuclei in the G2 and M phases of the cell cycle with double the amount of DNA. (B) A plot visualizing the internal complexity (SSC-A) and size (FSC-A) of all measured particles. The colors of the gating strategy represented in A are maintained in this plot. Single nuclei are red, aggregating nuclei are blue (located in gate A, but not in gate B), and single and aggregating debris (not present in gates A or B) are black. The single nuclei have low FSC-A and SSC-A values due to their small size and internal complexity and are mainly located in the bottom left corner of the plot. However, they considerably overlap with single debris particles, which also have a small size and internal complexity. This is the reason why this plot is not included in the gating strategy. Aggregating debris and nuclei (black) have a larger size and complexity resulting in increasing FSC-A and SSC-A values, respectively. (C) PI-A-based histogram representing a culture with three karyotype polymorphisms (1-3) present. The presence of one or two additional large chromosomes increases the amount of DNA in diploid nuclei, causing a clear shift in total PI fluorescence, resulting in this peak pattern.

Biology Open

using pulled glass pipettes. A mixture of 1:1 ethanol:glacial acetic acid was added dropwise to the cells, followed by pure glacial acetic acid. The slide with the fixed material was dried at 60°C and stained using Vectashield with DAPI (Vector Labs, H-1200). Images were captured using a Zeiss Axio Scope A1 microscope with an MRc5 digital camera.

## Acknowledgements
We thank Kirill Ustyantsev for providing feedback on the manuscript and Frank Beltman for helping with metaphase chromosome preparations.

## Competing interests
The authors declare no competing or financial interests.

## Author contributions
Conceptualization: S.M., E.B.; Funding Acquisition: E.B.; Experiments: S.M., L.G.; writing: S.M., E.B.

## Funding
This work was supported by University of Groningen core funding to E.B. Open Access funding provided by University of Groningen. Deposited in PMC for immediate release.

## Data and resource availability
All relevant data and details of the resources can be found in this article and supplementary figures.

## Peer review history
The peer review history is available online at https://journals.biologists.com/bio/lookup/doi/10.1242/bio.062346.reviewer-comments.pdf

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
