## [Peer Review File · Biology Open]

Tracking karyotype dynamics by flow cytometry reveals de novo chromosome duplications in laboratory cultures of *Macrostomum lignano*

Lisa Glazenburg, Eugene Berezikov and Stijn Mouton
DOI: 10.1242/bio.062346

Editor: Alissa Armstrong

Review timeline

Original submission:	28 October 2025
Editorial decision:	7 November 2025
First revision received:	19 December 2025
Accepted:	23 December 2025

Original submission

First decision letter

MS ID#: bio.062346

MS Title: Tracking karyotype dynamics by flow cytometry reveals de novo chromosome duplications in laboratory cultures of *Macrostomum lignano*

Authors: Stijn Mouton; Lisa Glazenburg; Eugene Berezikov

I have now reached a decision on the above manuscript.

The reviewer reports are shown at the bottom of this email.

As you will see, the reviewers gave quite favourable reports, but raised some minor points that will require amendments, mainly to the text, to your manuscript. I hope that you will be able to carry these out, because we would like to be able to accept your paper.

At this stage, we also ask you to ensure your manuscript complies with our formatting guidelines - please see our manuscript preparation guidelines for details. Provided you are able to fully address the referees' comments, we are positive about publication of your paper (we accept over 95% of revision submissions) and therefore hope you won't mind any extra work involved in reformatting your manuscript at this point.

Please upload both a 'clean' version of your Word file, along with a highlighted version clearly showing where you have made changes in the revised manuscript. Please avoid using 'Track changes' in Word files as these are lost in PDF conversion.

I should be grateful if you would also provide a point-by-point response detailing how you have dealt with the points raised by the reviewers in the 'Response to Reviewers' box. Please attend to all of the reviewers' comments. If you do not agree with any of their criticisms or suggestions please explain clearly why this is so.

Reviewer 1

Comments for the author

Mouton et al. manuscript titled "Tracking karyotype dynamics by flow cytometry reveals de novo chromosome duplications in laboratory cultures of *Macrostomum lignano*" describes the duplication events observed for the two large chromosomes of the flatworm *Macrostomum lignano* during its laboratory cultivation over a period of 26 months. The manuscript further establishes the flow cytometry method to rapidly test *M. lignano* cultures and highlights the importance of such testing in maintaining a culture with a specific karyotype. Finally, during their long-term experiment, the authors observed that the karyotype changes in *M. lignano* are unidirectional and, at least under laboratory conditions, are beneficial.

In summary, this manuscript provides valuable observations and establishes a simple methodology that will benefit the research community working with *M. lignano* and help guide future studies with this model organism.

Specific comments;

1- Introduction line 86, sentence starting with "This experiment was performed..." All the text following this sentence should be removed or moved to the results section, or rewritten to maintain the flow of the introduction without abruptly jumping into results.

2- Considering the flow cytometry methodology in this manuscript is central to its main message, this section requires clarification and additional details to enable others to perform similar experiments. For example, the single-nuclei preparation section should specify how many times one should pipette to obtain smaller fragments, what type of pipette should be used, whether they are pipetting gently or harshly and what "small" means. Next, it should describe how the fragments were dissociated into single cells by pipetting, including the duration, the type of pipette and tip used, and how to determine if the process has been effective. Finally, once the cells are in the nuclei isolation buffer, are they incubated or passed directly through a filter? Which filter is used? Is a syringe employed? How do you evaluate whether the protocol has been successful?

3- The flow cytometry steps could be described more clearly. How were the gating values decided? It is clear to see in the flow cytometry or does it require running control sets in advance? Can the authors include a sample data sheet to help people observe the gating values?

4- Figure 3B needs further explanation for readers less familiar with flow cytometry.

5- The authors present an example of karyotyping metaphase chromosomes in Figure 1B. Could they verify any of the subcultures with different karyotypes using the same methodology to show that both methods can yield similar ratios? If this has already been demonstrated in a previous study, it should be clarified in the text.

Reviewer 2

Comments for the author

The manuscript describes a method for rapidly obtaining information on DNA content as a proxy of the putative karyotype of worms from the *Macrostomum lignano* species, which is known to have a unique chromosomal features arising from a recent genome duplication event.

The authors provide a nice review of the state of the art in the introduction in the context of this species, their known karyotypes and they cite all relevant work of other labs, including previous work by them.

The method proposed has some bench procedures that have to be performed but would otherwise be a quick way to assess the DNA content of a culture of worms, changes of DNA content over time, etc. and hence represents a nice addition to the toolset available for these (and maybe also other) similar species.

The manuscript is well-written, down to the point and I have not found any red flags. Congrats to the authors.

One thing that may be improved is perhaps the summarization of all the cytometry data. Instead of showing all the plots (as in the supp figures) for subcultures or maybe in addition to showing all plots, maybe also provide a schematic visualization of the path of chromosomal changes, number of subcultures where change has occurred. Or maybe as a table.

e.g.

subculture 1: initial $2n=8$  ...  ...  $2n=8$ (no change)

subcultures 2,7: initial $2n=8$  $2n=9$...  $2n=9$

subcultures 12: initial $2n=8$  ... 16 months  $2n=9$

Reviewer's Responses to Questions

Experimental quality

Does each figure have the proper controls?

If 'No', please indicate reasons in Comments for Author box below.

Reviewer #1:

- Yes

Reviewer #2:

- Yes

Were the data analyzed using appropriate statistical tests?

If 'No', please indicate reasons in Comments for Author box below.

Reviewer #1:

- Yes

Reviewer #2:

- Yes

Reproducibility

Were experiments performed using adequate number of biological replicates?

If 'No', please indicate reasons in Comments for Author box below.

Reviewer #1:

- Yes

Reviewer #2:

- Yes

Does the methods section provide sufficient detail to permit reproducibility?

If 'No', please indicate reasons in Comments for Author box below.

Reviewer #1:

- No

Reviewer #2:

- Yes

Completeness

Are the manuscript's conclusions supported by the data?

If 'No', please indicate reasons in Comments for Author box below.

Reviewer #1:

- Yes

Reviewer #2:

- Yes

Scholarship

Do the authors cite and discuss the merits of data that would argue for and against their conclusion?

If 'No', please indicate reasons in Comments for Author box below.

Reviewer #1:

- Yes

Reviewer #2:

- Yes

Does the manuscript title & abstract accurately reflect the contents of the manuscript, without hyperbole?

If 'No', please indicate reasons in Comments for Author box below.

Reviewer #1:

- Yes

Reviewer #2:

- Yes

First revision

Author response to reviewers' comments

Response to reviewers

We thank the editor and reviewers for their appreciation of our research and their constructive feedback. We followed most of the suggestions. The text in the manuscript has been adjusted, and a point-by-point response can be found below.

Point-by-point response

To Reviewer 1:

1- Introduction line 86, sentence starting with "This experiment was performed..." All the text

following this sentence should be removed or moved to the results section, or rewritten to maintain the flow of the introduction without abruptly jumping into results.

- (1) This text has now been removed to avoid an abrupt transition to the results section, which has improved the flow of the manuscript.

2- Considering the flow cytometry methodology in this manuscript is central to its main message, this section requires clarification and additional details to enable others to perform similar experiments. For example, the single-nuclei preparation section should specify how many times one should pipette to obtain smaller fragments, what type of pipette should be used, whether they are pipetting gently or harshly and what "small" means. Next, it should describe how the fragments were dissociated into single cells by pipetting, including the duration, the type of pipette and tip used, and how to determine if the process has been effective. Finally, once the cells are in the nuclei isolation buffer, are they incubated or passed directly through a filter? Which filter is used? Is a syringe employed? How do you evaluate whether the protocol has been successful?

- (2) We have expanded this section to further clarify the method and provide additional details, as suggested. We hope that this will enable others to perform these experiments.
We further want to point out that the method is very robust and rather intuitive after performing it at least once. To evaluate whether the protocol was successful, we only looked at the results of the flow cytometry while running the samples, as this made things very clear very quickly.

3- The flow cytometry steps could be described more clearly. How were the gating values decided? It is clear to see in the flow cytometry or does it require running control sets in advance? Can the authors include a sample data sheet to help people observe the gating values?

- (3) We elaborated on the steps of the flow cytometry analysis.
For clarity, we here want to further point out that the gating is not based on specific values, but rather on the visual interpretation of the measured nuclei and debris. For the first experiments, it is recommended to run non-labelled control samples to be able to identify PI-positive nuclei. However, once some experience is obtained, this is even easy without running the controls. It is also important to realize that flow cytometry is not an inherently absolute technique. The laser power can be adjusted to change the location of the measured events in the graphs, and there will be some level of variation (e.g., the number of nuclei aggregates) between different samples. However, patterns will be relatively constant between samples, making the recognition of nuclei versus debris and interpretation of karyotype polymorphisms straightforward.
To further help the readers observe the location of the gates, we added a print screen of an example Kaluza data sheet as Supplementary Figure 15.

4- Figure 3B needs further explanation for readers less familiar with flow cytometry.

- (4) We have rephrased the legend of Figure 3B to explain it better to readers who are less familiar with flow cytometry.

5- The authors present an example of karyotyping metaphase chromosomes in Figure 1B. Could they verify any of the subcultures with different karyotypes using the same methodology to show that both methods can yield similar ratios? If this has already been demonstrated in a previous study, it should be clarified in the text.

- (5) After the 26 months-long experiment finished, we did not keep the subcultures with multiple karyotypes, as our general aim is to maintain cultures without chromosome duplications. Therefore, the requested experiments could not be performed.

To Reviewer 2:

1- One thing that may be improved is perhaps the summarization of all the cytometry data. Instead of showing all the plots (as in the supp figures) for subcultures or maybe in addition to showing all plots, maybe also provide a schematic visualization of the path of chromosomal changes, number of subcultures where change has occurred. Or maybe as a table.

- (1) In addition to the plots in the supplementary figures, we added a schematic visualization of the timeline of chromosome changes in the 12 subcultures as Figure 1 G.

Second decision letter

MS ID#: bio.062346R1

MS TITLE: Tracking karyotype dynamics by flow cytometry reveals de novo chromosome duplications in laboratory cultures of *Macrostomum lignano*

AUTHORS: Stijn Mouton; Lisa Glazenburg; Eugene Berezikov

I am happy to tell you that your manuscript has been accepted for publication in Biology Open, pending our standard publication integrity checks. It was accepted on 23rd December 2025.